# Deceptive Tricks in Artificial Intelligence: Adversarial Attacks in Ophthalmology

**DOI:** 10.3390/jcm12093266

**Published:** 2023-05-04

**Authors:** Agnieszka M. Zbrzezny, Andrzej E. Grzybowski

**Affiliations:** 1Faculty of Mathematics and Computer Science, University of Warmia and Mazury, 10-710 Olsztyn, Poland; 2Faculty of Design, SWPS University of Social Sciences and Humanities, Chodakowska 19/31, 03-815 Warsaw, Poland; 3Institute for Research in Ophthalmology, Foundation for Ophthalmology Development, 60-836 Poznan, Poland

**Keywords:** adversarial attacks, ophthalmology, artificial intelligence

## Abstract

The artificial intelligence (AI) systems used for diagnosing ophthalmic diseases have significantly progressed in recent years. The diagnosis of difficult eye conditions, such as cataracts, diabetic retinopathy, age-related macular degeneration, glaucoma, and retinopathy of prematurity, has become significantly less complicated as a result of the development of AI algorithms, which are currently on par with ophthalmologists in terms of their level of effectiveness. However, in the context of building AI systems for medical applications such as identifying eye diseases, addressing the challenges of safety and trustworthiness is paramount, including the emerging threat of adversarial attacks. Research has increasingly focused on understanding and mitigating these attacks, with numerous articles discussing this topic in recent years. As a starting point for our discussion, we used the paper by Ma et al. “Understanding Adversarial Attacks on Deep Learning Based Medical Image Analysis Systems”. A literature review was performed for this study, which included a thorough search of open-access research papers using online sources (PubMed and Google). The research provides examples of unique attack strategies for medical images. Unfortunately, unique algorithms for attacks on the various ophthalmic image types have yet to be developed. It is a task that needs to be performed. As a result, it is necessary to build algorithms that validate the computation and explain the findings of artificial intelligence models. In this article, we focus on adversarial attacks, one of the most well-known attack methods, which provide evidence (i.e., adversarial examples) of the lack of resilience of decision models that do not include provable guarantees. Adversarial attacks have the potential to provide inaccurate findings in deep learning systems and can have catastrophic effects in the healthcare industry, such as healthcare financing fraud and wrong diagnosis.

## 1. Introduction

In many areas of artificial intelligence, machine learning algorithms based on big data (Wang et al. [1] and Ching-Yu et al. [2]) and deep learning have facilitated extraordinary progress, with many applications in ophthalmology (Keenan et al. [3], Papadopoulos et al. [4], Rampasek et al. [5], Ishii et al. [6], Kermany et al. [7], Liu Y. et al. [8], Liu T. et al. [9], Burlina et al. [10], Cen et al. [11], Zheng et al. [12], Shekar et al. [13], and by Zhao et al. [14]).

Advances in artificial intelligence (AI) algorithms and easy access to large public datasets have made AI models ubiquitous, from funny filters on Instagram to automatic translators that help us perform our work to specialised models that solve complicated problems such as analysing and classifying medical photos.

With the help of artificial intelligence algorithms, we can solve complex image classification problems in different areas of life. The development of big data has given us access to millions of correctly classified images. This allows the algorithms to learn better. The latest models can achieve up to 98% accuracy, which can be misleading. Suppose we consider even minor perturbations to the image, such as the change in colour of just one pixel. Then, such models are uncertain for small perturbations. A disturbance of the image, which does not change in termsof human perception, makes it a completely different image for the AI model (Figure 1).

The figure above demonstrates that the model first correctly categorised the panda image. After the perturbation was applied to this image, the model’s prediction shifted to a gibbon, and the likelihood that it was a gibbon was extremely high. Humans interpret both photographs similarly. Nonetheless, the model views them as entirely distinct. This highlights the significance of adversarial attacks and that even modifications in the image that are imperceptible to humans can dramatically misrepresent the model’s output, as described by Liu et al. [16].

Most of the attacks presented in this paper concern convolutional neural networks (CNNs). However, the relatively new approach in computer vision, adapted from natural language processing, vision transformers (ViTs), can achieve state-of-the-art or near-state-of-the-art performance on various image classification tasks, as demonstrated by recent advances in attention-based networks. It distinguishes transformers as a promising alternative to conventional convolutional neural networks. Mahmood et al. [17] demonstrated that white-box attacks are highly effective at producing vision transformer examples. They concluded that vision transformers are as susceptible to white-box adversaries as their CNN counterparts. Nonetheless, transferability can be utilised to achieve robustness against a black-box adversary. They developed a SAGA attack.

Hu et al. [18] also employed attacks against CNNs and ViTs. They described the IAM-UAP attack algorithm.

However, some groundbreaking studies by Naseer et al. [19] demonstrated that ViTs are more robust than CNNs against adversarial patch attacks, arguing that the dynamic, receptive field of multihead self-attention (MSA) is the reason for its superior robustness.

Wang and Ruan provided significant results [20]. The authors demonstrated that ViTs are Lipschitz continuous for vision tasks and then formally connect the local robustness of transformers to the Cauchy problem. They theoretically showed that the maximum singular value determines the local robustness of each block’s Jacobian. They discovered that the initial and final layers inhibit the robustness of ViTs. In addition, contrary to existing research that suggests MSA can increase robustness, they found that the defensive power of MSA in ViT is only effective against weak adversarial attacks for large models. MSA compromises adversary robustness even against solid attacks.

Unfortunately, research using ViT to classify medical images is relatively new, and there are no presented attacks on such models for classifying medical images. It could be an interesting step in the development of specialised adversarial attacks.

In recent years, more research has been undertaken on applying artificial intelligence algorithms to solve medical problems, i.e., studies on various eye diseases. The resulting models are highly efficient. Nevertheless, we typically do not receive any information on model checking. One such difficulty may arise from the adversarial attacks presented by Goodfellow et al. [15]. They are relatively easy attacks to develop and are expected when working with images.

This study covered adversarial and specific adversarial attacks for images used in ophthalmology. In this paper, we highlight the different attacks and describe their influence on models that receive ophthalmic images as input. We also address the necessity of designing novel attack algorithms for specific sorts of pictures, such as fundus or anterior segment images.

## 2. Adversarial Attacks

The tendency of classifiers to overfit has led many to assume that adversarial attacks only occur in deep neural networks. All decision models have an adversarial attack as a fundamental property. Here we will explore the fundamentals of adversarial attacks to better understand attacks on ophthalmic image classification models.

Suppose *d* represents the number of dimensions of the input object. Consider a data point representing an image $x_{0}\in R^{d}$ of class $C_{i}$, where 0<i≤j, and *j* is the number of classes. An adversarial attack is a malicious attempt to insert $x_{0}$ into a new data point $x\in R^{d}$ such that the classifier misclassifies *x*, i.e., *x* belongs to a particularly hostile target class.

There are two categories of attacks: targeted and untargeted attacks. Let us consider a binary classification. In a targeted attack, a certain class C_1_ is given, and with this attack, we want a given model M to classify an image of a given image I into class C_1_, although its correct class is C_2_. An untargeted attack aims to deceive the model M by giving it a distorted image I so that it is assigned to a class other than C_1_. Untargeted attacks are not as good as targeted attacks and take much less time. Targeted attacks are more effective at changing the model’s predictions, but they come at a price: time.

### Problem Formulation

An adversarial attack is a malicious attempt to perturb a data point $x_{0}\in R^{d}$ to another point $x\in R^{d}$ so that *x* belongs to a particular adversarial target class.

Suppose a binary classifier classifies photographs of the anterior segment of the eye for cataracts.

For example, if $x_{0}$ is a feature vector of an anterior eye segment photograph with a correct classification of cataract. With an adversarial attack, we want to create another feature vector *x* classified as healthy (or another class specified by the attacker). In some scenarios, the goal may not be to assign $x_{0}$ to a particular target class C${}_{t}$, but to push it away from its original class C${}_{i}$.

Let us define a data set as $D={\left\{x_{i},y_{i}\right\}}_{i=1}^{N}$, where $x_{i}$ is a data sample labelled $y_{i}$, and *N* is the size of the data set. A trained model is denoted as M with input *x* and prediction M(x). A loss function is denoted by *ℓ*. In a soft machine learning system, training attempts to minimise the loss between the target label and the predicted label:

**Definition** **1.**
*Given data $X=\{x_{1},x_{2},\ldots ,x_{n}\}$ and target labels $Y=\{y_{1},y_{2},$$\ldots ,y_{n}\}$ find a hypothesis H such that*

$$\underset{H}{arg min}\sum_{x_{i},y_{i}\in D}\ell \left(H\left(x_{i}\right),y_{i}\right).$$



The trained model is tested to see how well it can predict the predicted label. One of the methods is to calculate the error by summing the losses between the target label and the predicted label.

Let us define a test dataset as $D^{\prime }={\left\{x_{i},y_{i}\right\}}_{i=1}^{M}$, where $x_{i}$ is a data sample with label $y_{i}$, and *M* is the size of the dataset.

**Definition** **2.**
*Given test data $X^{\prime }=\left\{x_{1},x_{2},\ldots ,x_{m}\right\}$ and test labels $Y^{\prime }=\left\{y_{1},y_{2},\ldots ,y_{m}\right\}$, the error is*

$$\sum_{x_{i},y_{i}\in D^{\prime }}\ell \left(H\left(x_{i}\right),y_{i}\right).$$



The adversarial attack algorithm is divided into two main parts:The query input is changed from the benign input *x* to $x^{\prime }$.An attack target is set so that the prediction result H(x) is no longer *y*. The loss is changed from $\ell (H\left(x_{i}\right),y_{i})$ to $\ell (H\left(x_{i}\right),y_{i}^{\prime })$, where $y_{i}^{\prime }\neq y_{i}$.

**Definition** **3** (Adversarial Attack (AA))**.**
*Let $x_{0}\in R^{d}$ be a data point belonging to class C${}_{i}$. Define a target class C${}_{t}$. An adversarial attack is a mapping $A:R^{d}\to R^{d}$ such that the perturbed data*

$$x=A(x_{0})$$



## 3. Methodology

As a starting point for our discussion, we used the study by Ma et al. [21].

We began our search of open-access research papers available in Google Scholar’s primary search. We searched for papers that consisted of entering various terms in three types of search engines, including “ophthalmology”, “ophthalmic image”, “retina”, and “retinal image”, “fundus”, and “fundus image”, along with the words “adversarial attacks” and “attacks”. The exact search was performed in the Google search and PubMed engines. We considered papers published after 2018.

## 4. Challenges in Adversarial Attacks in Medicine

Before the paper by Ma et al. [21], adversarial machine learning analysis focused on natural images, and medical image AAs were still unclear. Unlike natural images, medical images may contain domain-specific features. AAs can affect medical deep-learning systems. AAs can arbitrarily change diagnoses and outcomes. Ma et al. emphasised the importance of the fact that the healthcare system is heavily resourced. It inevitably creates risks where potential attackers can attempt to profit from manipulating the healthcare system. They gave an example of an attacker who can manipulate the health system’s investigative reports to commit insurance fraud or submit a fraudulent claim for medical reimbursement, as first presented by Paschali et al. [22].

In addition, an attacker may attempt to make a misdiagnosis by manipulating an image without detection. It may significantly affect patient decisions. DNNs operate on the black box principle, so such misclassification would be nearly impossible to detect. Other techniques can explain the DNNs’ decision, but consulting an expert/doctor is necessary for the diagnosis to be certain. This was shown by Avramidis et al. [23], Kind et al. [24], Chetoui et al. [25], and Ribeiro et al. [26].

Decision models and medical imaging techniques are increasingly used in medical diagnostics. Safe and robust medical deep-learning systems have become essential, as shown by by Finlayson et al. [27] and Paschali et al. [22]. The work by Ma et al. [21] was an essential step toward developing a comprehensive understanding of AAs in this field. It provides a broad knowledge of AAs in medical images, from the perspectives of both generating and detecting these attacks. However, Finlayson et al. [27] and Paschali et al. [22] have studied AAs on medical images, primarily focusing on testing the robustness of the deep models developed for medical image analysis.

The authors emphasised that AAs can succeed faster on medical images than natural ones. It means that fewer perturbations are required to execute a successful attack. They then pointed to two main reasons why deep neural networks that classify medical images are more exposed to AAs:Typically, medical images have complex biological textures, leading to areas with stronger gradients that are sensitive to small perturbations from attackers.State-of-the-art deep neural networks designed for large-scale natural image processing can be reparameterised for medical imaging tasks, resulting in a severe loss of landscape and high susceptibility to AAs.

According to the analysis by Ma et al. [21], medical imaging AAs generated using attack methods developed from natural images are not really “detrimental” in the medical sense. The authors noted that caution should be used when using these AAs to evaluate the performance of DNN models for medical images. Their study also shed light on the future development of more effective attacks on medical images. Future research should focus on developing specific AAs for different types of ophthalmic images.

### The Danger of Adversarial Attacks in the Healthcare System

According to Finlayson et al. [28], the U.S. healthcare system favours AAs. They summarised parts of the healthcare system that can give an attacker a reason and opportunity to perform an AA.

The paper also discusses the technical reasons why medical machine-learning systems are accessible to AAs. The authors noted that the truth must often be clarified and id more contentious. End users change images that are difficult to diagnose. In this case, they can make it very hard for medical experts to figure out how much influence they have. Deep learning is most useful in edge cases.

They also noted that medical imaging is highly standardised. They found that medical AA attacks do not require invariance. Medical AA attacks are not immune to lighting or position changes.

They also considered the fact that standard network architectures are frequently used. Most of the best-published methods in medical computer vision have the same basic structure. This lack of architectural variety makes it easier for attackers to attack multiple medical systems. Models should be made public to provide transparency and enable more targeted attacks.

Finlayson et al. [28] provided a hypothetical but intriguing example of adversarial examples in ophthalmology. According to the authors, government agencies often issue diagnostic guidelines requiring coverage for specific procedures, if requirements are met. One of the criteria could be that a patient with diabetic retinopathy should undergo a specific surgery from an insurance company. Even if the insurer cannot control the policy, it can control surgery frequency using noise instead of slightly positive photos. On the other hand, an ophthalmologist may place a universal enemy image on the lens of its image capture system. A third-party image processing system would mistakenly label all images as positive cases without changing the image in the information system.

## 5. Adversarial Attacks in Ophthalmology

Artificial intelligence has a significant role to play in the field of ophthalmic imaging. Colour fundus imaging and optical coherence tomography play a substantial role in ophthalmology in terms of their diagnostic accuracy and usefulness in segmenting disorders.

This section provides a summary of AAs on ophthalmology-related models. However, the original issue is partitioned into two separate ones. The first is the usage of classical adversarial algorithms, and the second, given in the following section, is the application of ophthalmology-specific adversarial algorithms. In chronological order, we began our examination in 2018.

Shah et al. [29] studied the effect of AAs on retinal imaging. To detect diabetic retinopathy, they evaluated image-based CNN-0, closely influenced by Alexnet by Krizhevsky et al. [30]; CNN-1, inspired by Enet by Paszke et al. [31]; and hybrid-lesion-based medical image analysis models by Abràmoff et al. [32]. The CNN-1 and hybrid-lesion-based models were evaluated using CNN-0 and the iterative fast gradient sign method (I-FGSM) by Kurakin et al. [33] to produce adversarial images. According to the experimental results, CNN models performed rather poorly, but hybrid-lesion-based models were more robust and achieved 45% and 0.6% reductions in accuracy, respectively (Figure 2). They collected the image from the Eyepacs dataset by Cuadros and Bresnick [34]. The authors noted that the explainability of the decision process underlying medical-image-based diagnostic tasks is of utmost importance.

Medical AAs were demonstrated to be feasible by Finlayson et al. [28]. The authors created a set of medical classifiers based on state-of-the-art clinical deep-learning algorithms. White- and black-box attacks against each other were the two types of AAs they carried out.

They created classification models for referable diabetic retinopathy based on fundoscopy images of the retina (similar to Gulshan et al. [35]). Data from the Kaggle Diabetic Retinopathy Dataset were used to train the models. Instead of the retinopathy grade, the Kaggle dataset sought to predict referable (grade 2 or worse) diabetic retinopathy using the model by Gulshan et al. [35].

Two different sorts of attacks, human undetected and patch attacks, were used to show how vulnerable their models were to AAs under various threat models. The white- and black-box projected gradient descent (PGD) attack methods were used in the undetectable by human attacks. The FGSM attack, first presented by Madry et al. [36], has evolved into the PGD attack, an iterative evolution of the FGSM attack. They adapted the approach outlined by Buckman et al. [37] for adversarial patch attacks.

The experimental results (Figure 3) showed that even very accurate medical classifiers can produce AAs, regardless of whether potential attackers have direct access to the model or must attack humans invisibly. PGD attacks require digital access to specific photographs provided to the model, while adversarial patch attacks can be used on any image, according to the research. They described realistic AAs against their systems.

Ma et al. [21] examined four attacks on DNNs trained on five medical image datasets. They presented attack parameters, results, and analyses. FGSM, BIM, PGD, and the strongest optimisation-based Carlini and Wagner attack ($L_{\infty }$ version) were performed. These attacks were limited by a maximum perturbation ϵ in the $L_{\infty }$ norm on each input pixel.

They benchmarked the difficulty of classifying diabetic retinopathy fundus images. They initially categorised the image dataset into ‘no DR’ and ‘referable’. They focused on medical photos’ AA difficulty, which matched ImageNet’s nature images. Medical photos were easier to attack than ImageNet photos, the findings showed. This unexpected finding was clarified by a saliency map for two ImageNet and medical images from different classes. Based on classification loss gradients, the saliency map of an input picture shows the regions that change model output most from classification loss gradients. However, comparing the model procedures with the explanations of ImageNet and medical image output from different classes would be interesting, valuable, and intriguing.

As a result, we concluded that we require explainable models in addition to unique AAs for characteristic medical images.

Yoo and Choi [38] investigated whether AAs can confound deep-learning systems based on imaging techniques such as fundus photography (FP), ultra-widefield photography (FP), and optical coherence tomography (OCT). Based on the basic FGSM algorithm, they developed a binary classifier to identify cases of diabetic retinopathy and diabetic macular oedema. Again, the unique qualities of medical imagery were ignored in this study.

The results exhibited the same characteristics as those reported by Ma et al. [21]. Using FGSM attacks helps deep learning models perform better.

Nonetheless, the authors adopted the same stance on attacks as Ma et al. [21]. Attackers may disrupt hospitals’ and insurance firms’ medical billing and reimbursement systems. Defensive approaches, including adversarial training, denoising filters, and generative adversarial networks, can successfully decrease the impact of AAs.

Lal et al. [39] worked on establishing a framework for diabetic retinopathy detection-specific attack detection and defence. Keeping algorithms safe and reliable is a major concern in the fast-paced growth of artificial intelligence (AI) and deep learning (DL) techniques. The main conclusion of the study was a framework that provides a protective model against the adversarial speckle-noise attack and adversarial training. The authors then proposed a feature fusion technique that maintains categorisation with correct labelling. They assessed and studied the AAs and defences created by their framework on retinal fundus images for the problem of diabetic retinopathy identification. However, the authors concentrated on only two types of attacks: FGSM and speckle-noise attack (SN). The most significant disadvantage is that these attacks are detectable at first glance. The authors correctly noted that medical images are significantly degraded because, for example, noise is unavoidable in data acquisition, contrast is low due to illumination variations, and various other factors can produce random pixel values for individual pixels in an image by multiplying speckle noise (Figure 4).

Hirano et al. [40] emphasised the need for more concentrated research on the vulnerability of DNNs to AAs. They focused on a single, modest, image-agnostic perturbation known as a universal adversarial perturbation (UAP), which can cause DNN failure in the majority of image classification tasks. Because such minute changes have little effect on data distributions, UAPs are challenging to find. Nonetheless, the authors found that UAP-based AAs are more accurate for adversaries to implement in real-world contexts. In medical imaging diagnostics, UAPs are prone to security risks.

Nonetheless, defence tactics against UAPs in DNN-based medical image classification still need to be studied, despite the sensitivity of DNNs to AAs, demonstrating the need for solutions to address security concerns.

Researchers focused on a representative medical image classification: the classification of diabetic retinopathy based on OCT images. Based on previous research, they built DNN models with different topologies for medical image recognition and investigated their susceptibility to untargeted and targeted attacks based on UAPs. They also studied the defence against attackers. They used adversarial training to test the increased resistance of DNNs to untargeted and targeted UAPs.

## 6. Specified Adversarial Attacks in Ophthalmology

The studies presented in the previous section have indicated a need to build algorithms for deception attacks explicitly targeted at medical images. In this section, we concentrate on attacks specifically designed for ophthalmic images in this area.

### 6.1. Adaptive Segmentation Mask Attack

Ozbulak et al. [41] demonstrated that deep-learning-based medical image segmentation models are susceptible to AAs. They concentrated on optic disc segmentation in glaucoma and targeted attacks. They employed the U-Net model, one of the best-recognised models for medical image segmentation. They introduced the adaptive mask segmentation attack (AMSA), a unique approach to creating AAs with realistic prediction masks. When misclassified, the algorithm modifies the masks in largely imperceptible ways to human sight. The authors posted the AA’s source code on GitHub at https://github.com/utkuozbulak/adaptive-segmentation-mask-attack/ (accessed on 25 April 2023).

### 6.2. HFC

Yao et al. [42] presented a novel hierarchical feature constraint (HFC) to supplement current white-box attacks, allowing the adversarial representation to be concealed within the normal feature distribution. They examined the proposed approach using a fundoscopy image dataset. Their investigations revealed that the increased susceptibility of medical representations gives an adversary more opportunities for malevolent manipulation. Their premise was that medical AAs are easily detectable. They specified the HFC that can be used in attacks to evade detection. They performed comprehensive tests to determine the efficacy of the HFC. The experimental results demonstrated that HFC significantly outperformed other adaptive attacks, highlighting the inadequacies of existing approaches for identifying medical attacks in feature space.

### 6.3. MSA

Shao et al. [43] reported a targeted attack against a biomedical image segmentation model built on multiscale gradients. Their approach integrates the attack on the adaptive segmentation mask with a perturbation of the feature space. They suggested a targeted attack approach employing multiscale gradients and applied the binary cross-entropy loss function for semantic image segmentation. The MSA approach introduces the gradient information of the loss functions to repeatedly calculate the adversarial perturbation. The original image is fitted with the adversarial perturbation and is subsequently transformed to an AA when the predicted mask of AA is close to the target mask. They conducted glaucoma-related studies utilising U-Net and an optic disc segmentation dataset. They demonstrated that their MSA method outperformed the standard method (ASMA).

### 6.4. SMIA

In order to diagnose pre-existing medical systems, Qi et al. [44] presented a method for medical image AA. By maximising the deviation loss term and minimising the loss stabilisation term, the approach iteratively builds AAs. The current iteration increases the difference between the CNN prediction and the matching ground truth label for AA. In comparison, the CNN predictions for the smoothed input are similar. Similar predictions ensure that the perturbations are created in a relatively flat region in the CNN feature space. The perturbations are updated from one flat region to another throughout subsequent iterations. Hence, the suggested approach cab explore perturbation space to smooth a single point and obtain a local optimum. Compared with the perturbation movement between single locations induced just by the loss deviation term, the loss stabilisation term enhances attack performance by stabilising the perturbation movement.

They used two datasets for diabetic retinopathy evaluation: APTOS-2019 and Kaggle-DR, and they used the ResNet-50 model combined with a graph convolutional network to classify fundus images. Under the same medical analytic settings, they compared the proposed SMIA technique with existing AA methods. In addition to the commonly utilised FGSM and PGD algorithms, they included DAG, which was proposed for natural image identification and segmentation, and DeepFool, which was designed for disease classification. The visualisation and experimental results proved the attack approach’s success in determining the limitations of medical diagnostic systems and further enhancing them (Figure 5).

### 6.5. Remarks

The aforementioned studies have illustrated specific attack tactics for medical images. However, no novel algorithms for attacks on various ophthalmic image types have yet to be developed, which is a task that must be accomplished. As a result, algorithms that explain the results of artificial intelligence models and that validate the computation of artificial intelligence models must be developed.

## 7. Regulations

### 7.1. The European Union

In April 2021, the European Commission has proposed REGULATION OF THE EUROPEAN PARLIAMENT AND OF THE COUNCIL LAYING DOWN HARMONISED RULES ON ARTIFICIAL INTELLIGENCE (ARTIFICIAL INTELLIGENCE ACT) AND AMENDING CERTAIN UNION LEGISLATIVE ACTS [45].

Note 51 says:


*“Cybersecurity plays a crucial role in ensuring that AI systems are resilient against attempts to alter their use, behaviour, performance or compromise their security properties by malicious third parties exploiting the system’s vulnerabilities. Cyberattacks against AI systems can leverage AI specific assets, such as training data sets (e.g., data poisoning) or trained models (e.g., adversarial attacks), or exploit vulnerabilities in the AI system’s digital assets or the underlying ICT infrastructure. To ensure a level of cybersecurity appropriate to the risks, suitable measures should therefore be taken by the providers of high-risk AI systems, also taking into account as appropriate the underlying ICT infrastructure.”*


Chapter 2, Article 15, point 4 says:


*“High-risk AI systems shall be resilient as regards attempts by unauthorised third parties to alter their use or performance by exploiting the system vulnerabilities.*

*The technical solutions aimed at ensuring the cybersecurity of high-risk AI systems shall be appropriate to the relevant circumstances and the risks.*

*The technical solutions to address AI specific vulnerabilities shall include, where appropriate, measures to prevent and control for attacks trying to manipulate the training dataset (‘data poisoning’), inputs designed to cause the model to make a mistake (‘adversarial examples’), or model flaws.”*


### 7.2. The United States of America

In January 2021, the U.S. Food and Drug Administration (FDA) presented “Artificial Intelligence/Machine Learning (AI/ML)-Based Software as a Medical Device (SaMD) Action Plan” [46]. Point 4 “4. Regulatory Science Methods Related to Algorithm Bias & Robustness” says:


*“Support regulatory science efforts to develop methodology for the evaluation and improvement of machine learning algorithms, including for the identification and elimination of bias, and for the evaluation and promotion of algorithm robustness.”*


### 7.3. Discussion

We could not find real-life adversarial attacks in the medicine descriptions in the literature. This shows how complicated and delicate the problem is. In the previous sections, we showed that two famous institutions are aware of the possibility of attacks and the lack of robustness of the models.

During our previous research, we used several commercial models. We proposed the algorithms’ owners’ joint research and the possibility of testing models using attacks several times. We also explained to companies that successful attacks will not discredit their product but show how their model can be improved. None of the companies expressed willingness to cooperate.

Testing the models by sending perturbed images via the interface provided by the companies is possible. However, such attacking could be noticed by the fact that we would send a dozen or so, at first glance, photos for classification that did not differ in any way. This alsod involve high costs. The results obtained in this way could not be made available and published.

We are currently working on attacks against a model provided to us by another research group. However, again, this is not a commercially used model.

Soon, legal regulations requiring the verification of models and their explainability will come into force in the European Union. However, we should be aware that such a process may take a long time, and many misdiagnoses will occur before the law comes into force. Similar problems occurred with the rapid development of information systems in the 2000s. Many errors in the operation of the systems were detected after implementation, costing human lives and financial losses. Legal regulations regarding formal verification were introduced over time and in a few countries worldwide mainly because the cost of formal verification often exceeded the cost of software development.

## 8. Conclusions

Artificial intelligence algorithms may advance ophthalmic image processing and become a vital tool for clinicians and hospitals. However, AAs present a significant risk to patients as well as a barrier to the correct operation of AI models. The purpose of this study was to provide a summary of the studies focused on the use of AAs in ophthalmic image analysis. The analyses have demonstrated that these attacks can also mislead ophthalmic imaging models. Several researchers have presented novel attacks that have been created specifically for the field of ophthalmic image processing and have a very high level of accuracy. We provide a brief summary of the described attacks in Table 1 and Table 2.

We think that computer scientists and the ophthalmology communities should closely cooperate and concentrate on tackling AAs to integrate AI algorithms into real-world problems.

In future work, we would like to compare all these attacks on a common database to conclude which attack is the strongest and evaluate the detection/defence methods, i.e., adversarial training, or explain the model susceptible to attacks using explainable methods.

## Figures and Tables

**Figure 1 jcm-12-03266-f001:**
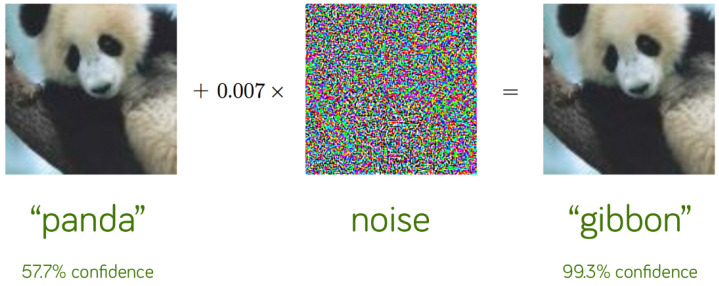
A classical example of an adversarial attack (Goodfellow et al. [15]).

**Figure 2 jcm-12-03266-f002:**
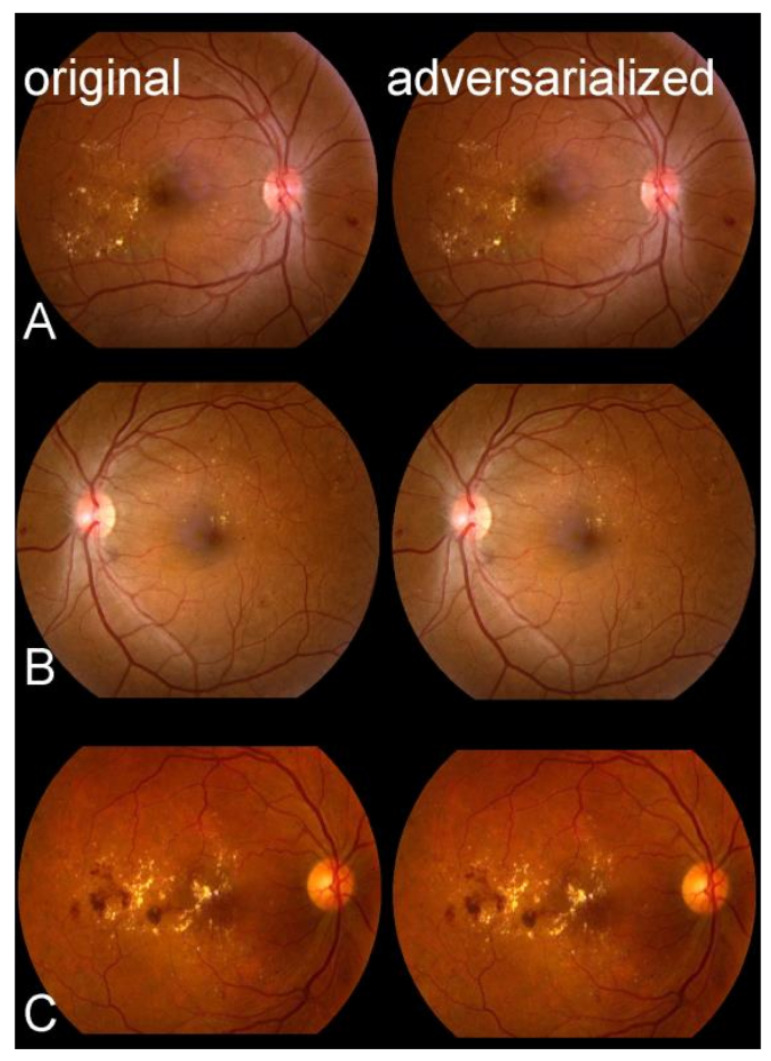
The AAs of three different patients (**A**–**C**) images that had previously been classified as a disease (referable diabetic retinopathy). The original images containing the disease are in the left column, while the right column displays the AA variants. Pixel differences are difficult to distinguish, even at high magnification (Shah et al. [29]).

**Figure 3 jcm-12-03266-f003:**
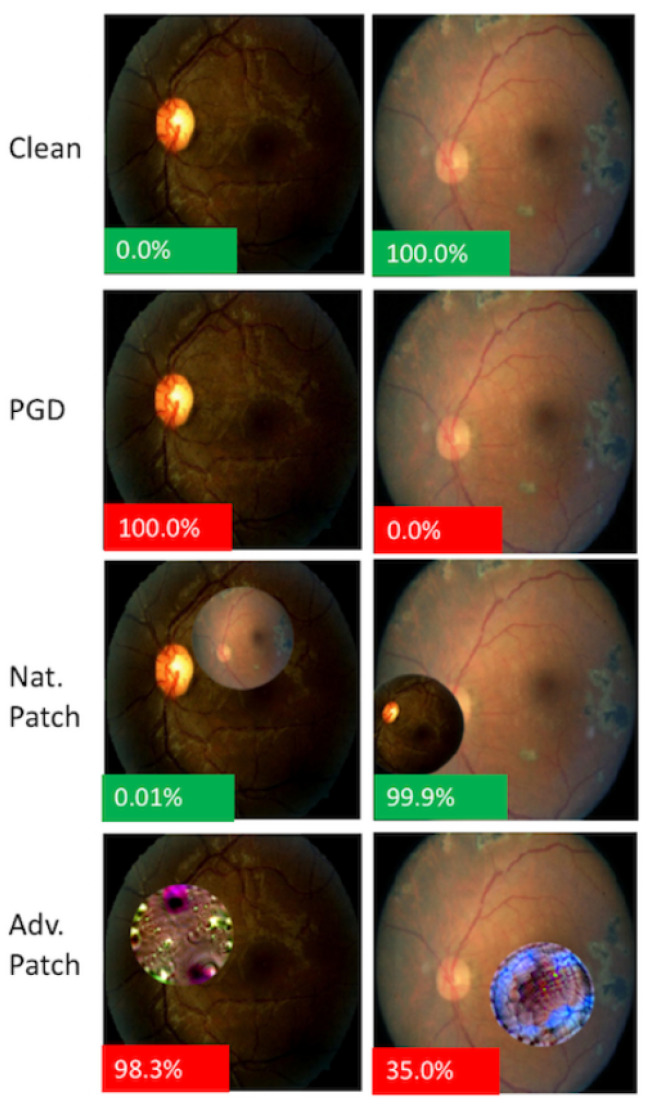
A demonstration that AAs can be feasible even for extremely accurate medical classifiers (Finlayson et al. [28]).

**Figure 4 jcm-12-03266-f004:**
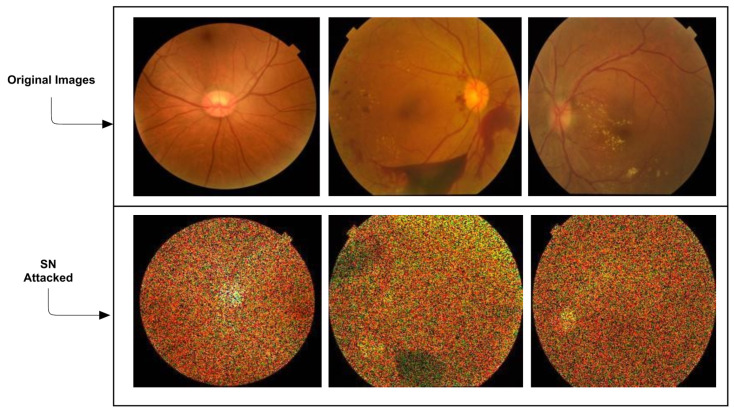
An example of the SN attacked images from [39] by Lal et al.

**Figure 5 jcm-12-03266-f005:**

An example of SMIA effectiveness (Qi et al. [44]).

**Table 1 jcm-12-03266-t001:** The standard AAs.

Type of the Attack	Characteristics
I-FGSM [29,33]	FGSM iteratively adds the noise (not random noise) whose direction is the same as the gradient of the cost function concerning the data.
PGD [28,36]	Initialises the attack to a random point in the ball of interest and performs random restarts.
FGSM [21,36,38]	Adds the noise (not random noise) whose direction is the same as the gradient of the cost function concerning the data.
BIM [21,33]	A simple extension of the FGSM where, rather than taking one significant step, it performs an iterative procedure by using FGSM numerous times on an image.
C&W [21,47]	A regularisation-based attack with some necessary modifications that can resolve the unboundedness issue.
SN [39]	The gritty salt-and-pepper pattern seen in radar imaging or a granular ‘noise’ that appears fundamentally in ultrasound.
UAP [40,48]	Leads to DNN failure in most image classification tasks.

**Table 2 jcm-12-03266-t002:** Specialised AAs.

Type of the Attack	Characteristics
AMSA [41]	Can produce AAs with realistic prediction masks.
HFC [42]	Enables hiding the adversarial representation in normal feature distribution.
SMIA [44]	The method iteratively generates AAs by maximising the deviation loss term and minimising the loss stabilisation term.
MSA [43]	Based on AMSA and multi-scale gradients.

## Data Availability

Data sharing not applicable No new data were created or analyzed in this study. Data sharing is not applicable to this article.

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
