# Peer review of "Deceptive Tricks in Artificial Intelligence: Adversarial Attacks in Ophthalmology"

_jcm, 2023, doi:10.3390/jcm12093266_

Round 1

Reviewer 1 Report

It's a very interesting research topic. The authors provided a proper review of this field. I thank the authors.

What should organizations like the FDA do about these attacks on AI? Do they need regulation and further evaluation? This article needs to address these arguments.

Are the latest VIT (vision transformer) models vulnerable to these attacks?

Reviewer 2 Report

The authors reviewed that the introduction of AI is progressing in ophthalmology, a clinical department with many images. On the other hand, while introducing AI is convenient, "Adversarial Attacks" confuse AI and confuse real-world medicine.

The review is clear, but if specific problems happened in medical departments are presented, even if not in ophthalmology fields we can have a greater sense of crisis about the incidents caused by "Adversarial Attacks."

Reviewer 3 Report

The title reads: “Adversarial Attacks in Ophthalmology”

Comment:

In artificial intelligence, adversarial attacks refer to techniques used to manipulate or deceive machine learning models by intentionally feeding them malicious inputs, known as adversarial examples, with the goal of causing the models to misclassify or make incorrect predictions. They pose a significant challenge to model robustness and security, and researchers are developing defenses to mitigate their impact. Adversarial attacks and defenses are ongoing areas of research in the field of artificial intelligence and machine learning, aimed at improving the security and robustness of machine learning models in real-world scenarios.  The title “Adversarial Attacks in Ophthalmology” is too succinct and does not place the reader in context. Someone who only reads the title might think that it is a study of ocular trauma in a competitive sports setting (boxing or judo), or in a setting of violence, which, of course, has nothing to do with it. Consider modifying to: “Deceptive Tricks in Artificial Intelligence: Adversarial Attacks in Ophthalmology".

Line 5. It reads: “However, there are several difficulties with the safety and trustworthiness of this technique, and tackling these issues is as vital as building AI systems to identify eye disease. Research has concentrated intensely on these difficulties, and numerous articles have appeared in recent years. We have used the paper [1] as a starting point for our discussion. Searches were performed in PubMed and Google search engines to identify the open-access research papers.”

Comment:

The paragraph is confusing, and does not mention the adversarial attacks.

Consider modifying to: “However, in the context of building AI systems for medical applications such as identifying eye disease, addressing the challenges of safety and trustworthiness is paramount, including the emerging threat of adversarial attacks. Research has increasingly focused on understanding and mitigating these attacks, with numerous articles discussing this topic in recent years [1]. A literature review was performed for this study, which included a thorough search of open-access research papers using online sources (PubMed and Google).”

Line 102. It reads: “As a starting point for our discussion, we have used the paper [1].”
Comment:

Consider modifying to: “As a starting point for our discussion, we have used the paper  by Ma et al. [1].”

Throughout the entire manuscript the authors used this way of referring to the articles by the citation number, but this way is not the conventional one, and it does not seem to be appropriate for a scientific text. They should then include the name of the first author, and add et al. if there are more than two authors, and then whether to add the reference between brackets. This adjustment must be made throughout the entire manuscript.

Table 1 and Table 2 are in the wrong position (in middle of the references list).

Round 2

Reviewer 2 Report

The authors reached the issue that I suggested before.